# Inhibitory Effect of Coumarins and Isocoumarins Isolated from the Stems and Branches of *Acer mono* Maxim. against *Escherichia coli β*-Glucuronidase

Nguyen Viet Phong [1,2], Byung Sun Min [3], Seo Young Yang [4,*] and Jeong Ah Kim [1,2,*]

[1] Vessel-Organ Interaction Research Center, VOICE (MRC), College of Pharmacy, Kyungpook National University, Daegu 41566, Korea
[2] BK21 FOUR Community-Based Intelligent Novel Drug Discovery Education Unit, College of Pharmacy and Research Institute of Pharmaceutical Sciences, Kyungpook National University, Daegu 41566, Korea
[3] Drug Research and Development Center, College of Pharmacy, Daegu Catholic University, Gyeongsan-si 38430, Korea
[4] Department of Pharmaceutical Engineering, Sangji University, Wonju 26339, Korea
* Correspondence: syyang@sangji.ac.kr (S.Y.Y.); jkim6923@knu.ac.kr (J.A.K.); Tel.: +82-33-738-7921 (S.Y.Y.); +82-53-950-8574 (J.A.K.)

**Abstract:** We isolated eight known secondary metabolites, including two isocoumarins and six coumarins, from the stems and branches of *Acer mono* Maxim. Their structures were confirmed using nuclear magnetic resonance spectroscopy and by comparing the data to published reports. The inhibitory effects of all compounds (**1**−**8**) on *Escherichia coli β*-glucuronidase were evaluated for the first time using in vitro assays. 3-(3,4-Dihydroxyphenyl)-8-hydroxyisocoumarin (**1**) displayed an inhibitory effect against *β*-glucuronidase ($IC_{50}$ = 58.83 ± 1.36 μM). According to the findings of kinetic studies, compound **1** could function as a non-competitive inhibitor. Molecular docking indicated that compound **1** binds to the allosteric binding site of *β*-glucuronidase, and the results corroborated those from kinetic studies. Furthermore, molecular dynamics simulations of compound **1** were performed to identify the behavioral and dynamic properties of the protein–ligand complex. Our results reveal that compound **1** could be a lead metabolite for designing new *β*-glucuronidase inhibitors.

**Keywords:** *Acer mono* Maxim.; isocoumarins; *β*-glucuronidase; non-competitive inhibitor; allosteric binding site; molecular dynamics

## 1. Introduction

*β*-Glucuronidase is a well-known enzyme that hydrolyzes conjugated compounds, including *β*-glucuronic acid, into their derivatives and free glucuronic acid [1]. *β*-Glucuronidase is frequently found in unicellular microorganisms, such as *Escherichia coli* and *Peptostreptococcus* species, and multicellular organisms, including plants and mammals [2,3]. In particular, this enzyme can be detected in several human body organs, such as the kidney, liver, lung, and digestive system [4]. In 2010, Wallace et al. first reported the crystal structure of *E. coli β*-glucuronidase [5]. The *β*-glucuronidase asymmetric unit (139 kDa) comprises two monomers (597 residues) and is organized into the following three areas: the N-terminal contains 180 residues and is similar to the carbohydrate-binding domain of the glycoside hydrolase 2 members; the C-terminal contains residue 274 to residue 603 and comprises an αβ loop; and the region between terminals N and C contains an Ig-like *β*-sandwich domain, as is the case with other members of the glycoside hydrolase 2 family [5,6]. Inhibiting *β*-glucuronidase can be beneficial for preventing and treating various diseases [7]. *β*-Glucuronidase is generated in the synovial fluid under inflammatory conditions, such as rheumatoid arthritis [8]. Moreover, an increased risk of colon cancer has been associated with the role of *β*-glucuronidase in the disease, as well as enhanced intestinal

enzyme levels [9]. Increased levels of $\beta$-glucuronidase in the blood, owing to liver injury, can lead to liver cancer [7,10]. Thus, the discovery and development of $\beta$-glucuronidase inhibitors may be valuable in reducing these carcinogenic risks.

*Acer* L., commonly known as maple, is a diverse genus in the Aceraceae family, which comprises over 125 species with several infraspecific taxa, and is widely distributed in northern temperate regions worldwide [11]. Many *Acer* species provide products of economic value, including furniture, lumber, horticultural plants, and herbal medicines, especially gamma-linolenic acid, a dietary supplement that can help treat cancer and cardiovascular diseases [12]. *A. mono* Maxim., a deciduous tree of the *Acer* genus, is commonly found in Korea, Japan, and Northeast China [13,14]. Over the years, this plant has attracted considerable attention from both chemists and pharmacists, given its noteworthy traditional uses and pharmacological activities. In Korea, the leaves of *A. mono* Maxim. have long been employed as a material for hemostasis, whereas the roots have been used in traditional Korean medicine to treat arthralgia and cataclasis [13–15]. The sap of *A. mono* Maxim. has been used to treat gout, neuralgia, urinary hesitancy, constipation, and other gastroenteric conditions [15,16]. *A. mono* Maxim. reportedly possesses various pharmacological properties, including hepatoprotective, antioxidant, and osteoporotic inhibitory effects [13,14]. Phytochemical constituents of *A. mono* Maxim. include stilbenes, flavonoids, and their derivatives [14]. Among them, 5-*O*-methyl-(*E*)-resveratrol 3-*O*-$\beta$-D-glucopyranoside and 5-*O*-methyl-(*E*)-resveratrol 3-*O*-$\beta$-D-apiofuranosyl-(1→6)-$\beta$-D-glucopyranoside, two stilbene glycosides isolated from the leaves of *A. mono* Maxim., could reduce the levels of DPPH radicals at the concentrations of 100 μM, thereby exhibiting significant free-radical scavenging effects, with $IC_{50}$ values of 103.6 and 80.5 μM, respectively [13].

As part of our ongoing investigation of the secondary metabolites and respective biological effects from herbal and medicinal plants in Korea, we isolated and structurally elucidated eight compounds, including two isocoumarins and six coumarins, from the stems and branches of *A. mono* Maxim. in the present study. To the best of our knowledge, this is the first report regarding the isolation of isocoumarins from the *Acer* genus. In vitro assays were performed to determine the $\beta$-glucuronidase inhibitory activity of all compounds isolated from *A. mono* Maxim. In addition, kinetic analysis studies, molecular docking, and molecular dynamics (MD) simulations were performed to comprehensively clarify the inhibition mode, critical amino acid interactions, and the protein–ligand binding mechanism of active compound **1** with $\beta$-glucuronidase proteins.

## 2. Results

### 2.1. Isolated Compounds from the Stems and Branches of A. mono Maxim.

Methanol residue from the stems and branches of *A. mono* Maxim. was suspended in distilled water and partitioned with *n*-hexane, $CH_2Cl_2$, and EtOAc to obtain four extracts. The $CH_2Cl_2$ extract (77.8 g) and water layer (170 g) were separated by repeated column chromatography (CC) on silica gel, RP-18 gel, and Diaion resins, followed by preparative high-performance liquid chromatography (HPLC) to isolate and purify two isocoumarins (**1** and **2**) and six coumarins (**3**−**8**) (Figure 1). Based on an in-depth analysis of nuclear magnetic resonance (NMR) results and comparison with corresponding data from previous reports, the structures of all the compounds isolated were elucidated as 3-(3,4-dihydroxyphenyl)-8-hydroxyisocoumarin (**1**) [17], hydrangenol (**2**) [18], scopoletin (**3**) [19], isoscopoletin (**4**) [19], isofraxidin (**5**) [20], fraxidin 8-*O*-$\beta$-D-glucopyranoside (**6**) [21], aquillochin (**7**) [22], and cleomiscosin D (**8**) [23].

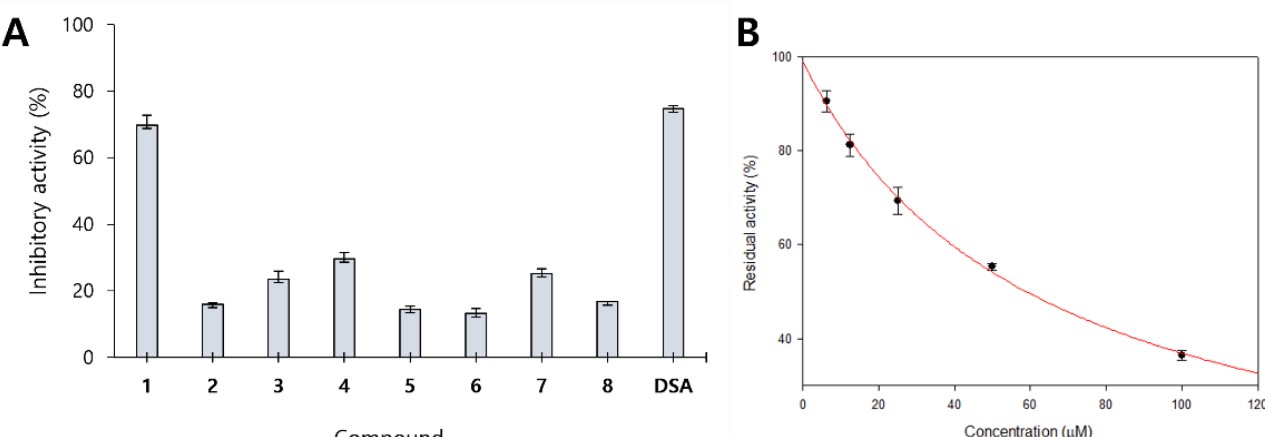

**Figure 1.** Chemical structures of isolated compounds from *Acer mono* Maxim. (**1**−**8**).

## 2.2. Inhibitory Activity of Isolated Compounds against β-Glucuronidase

All isolated compounds (**1**−**8**) were examined for their ability to inhibit β-glucuronidase using D-saccharic acid 1,4-lactone (DSA), a well-known inhibitor of β-glucuronidase, as a positive control [24]. The results are displayed in Figure 2A and Table 1, which revealed that 3-(3,4-dihydroxyphenyl)-8-hydroxyisocoumarin (**1**) exhibited β-glucuronidase inhibitory activity, with 69.85% of inhibition at 100 μM and an $IC_{50}$ value of 58.83 ± 1.36 μM. Compound **2** with the absence of the C-3−C-4 double bond in the lactone ring did not exhibit β-glucuronidase inhibitory activity (15.89% of inhibition at 100 μM). Coumarin and its derivatives (**3**−**8**) failed to exhibit inhibitory activity against β-glucuronidase ($IC_{50}$ >100 μM). These results suggested that the different positions of an oxygen atom and carbonyl group in the isocoumarin structure, compared with those of coumarin, could play an important role in β-glucuronidase inhibition.

**Figure 2.** Inhibitory activity of isolated compounds **1**−**8** at 100 μM (**A**) and determination of $IC_{50}$ value by compound **1** on β-glucuronidase (**B**).

**Table 1.** Inhibition of compounds **1–8** against $\beta$-glucuronidase.

| Compounds | Inhibition against $\beta$-Glucuronidase | |
|:---:|:---:|:---:|
| | Inhibition% (100 µM) | IC$_{50}$ (µM) [1] |
| **1** | 69.85 ± 2.93 | 58.83 ± 1.36 |
| **2** | 15.89 ± 0.55 | >100 |
| **3** | 23.46 ± 2.47 | >100 |
| **4** | 29.61 ± 1.83 | >100 |
| **5** | 14.34 ± 1.10 | >100 |
| **6** | 13.17 ± 1.46 | >100 |
| **7** | 25.21 ± 1.46 | >100 |
| **8** | 16.73 ± 0.09 | >100 |
| DSA [2] | 74.70 ± 0.95 | 24.56 ± 1.15 |

[1] The values (µM) represent 50% inhibition of $\beta$-glucuronidase. Results are presented as the mean ± standard error of triplicate experiments. [2] Positive control.

### 2.3. Enzyme Kinetics of Compound **1** against $\beta$-Glucuronidase

To determine the type of inhibition and the inhibitory constant $K_i$ of active compound **1**, enzyme kinetics were conducted at different concentrations of the substrate 4-nitrophenyl $\beta$-D-glucuronide (PNPG) and inhibitor [7]. In the Lineweaver−Burk plot, the x-axis is the reciprocal of the substrate concentration, or 1/(S), and the y-axis is the reciprocal of the reaction velocity (1/V). The non-competitive or competitive inhibition mode is indicated by the family of straight lines that intersect at the same point on the 1/(S) or 1/V axis, respectively, whereas mixed inhibition is represented by the straight lines of the inhibitor that intersect at the xy region. As depicted in Figure 3A, the Lineweaver−Burk plot revealed intersecting lines on the 1/(S) axis, indicating that compound **1** inhibited $\beta$-glucuronidase in a non-competitive inhibition mode. In addition, a Dixon plot was used to determine the $K_i$ value between the inhibitor and the enzyme, where the intersection value on the x-axis implies $K_i$. The $K_i$ is the concentration of an inhibitor needed to decrease the maximum rate of the reaction by 50% [25]. As presented in Figure 3B, the $K_i$ value of **1** was calculated to be 40.9 µM.

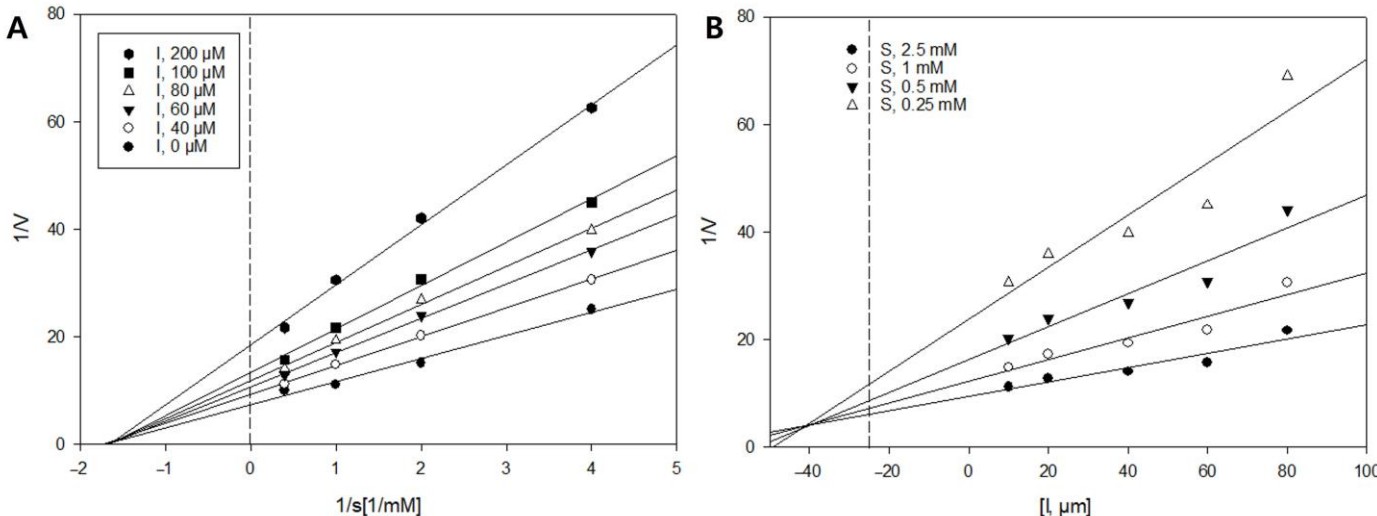

**Figure 3.** Lineweaver−Burk plot (**A**) and Dixon plot (**B**) analyses using active compound **1**.

### 2.4. Molecular Docking Studies

Molecular docking simulations were performed using AutoDock 4.2 to predict how compound **1** behaves in the binding site of the target protein and to clarify the fundamental biochemical processes of **1** with the $\beta$-glucuronidase enzyme. The results were analyzed and displayed using PyMOL and BIOVIA Discovery Studio (Figure 4). According to

the kinetic analysis results, compound **1** displayed a non-competitive inhibition mode, suggesting that **1** could precisely bind to a specific region of β-glucuronidase, named the allosteric binding site, which differed from the active site (orange spheres) that binds to the substrate PNPG (red). The allosteric binding site of β-glucuronidase was predicted using the AlloSite 2.10 web server (green spheres) [26] and protein allosteric sites server PASSer2.0 (rainbow spheres) [27] (Figure 4A). To optimize the docking procedure, the substrate, PNPG, was docked as a native ligand into β-glucuronidase (PDB ID: 6LEL) (Figure 4B).

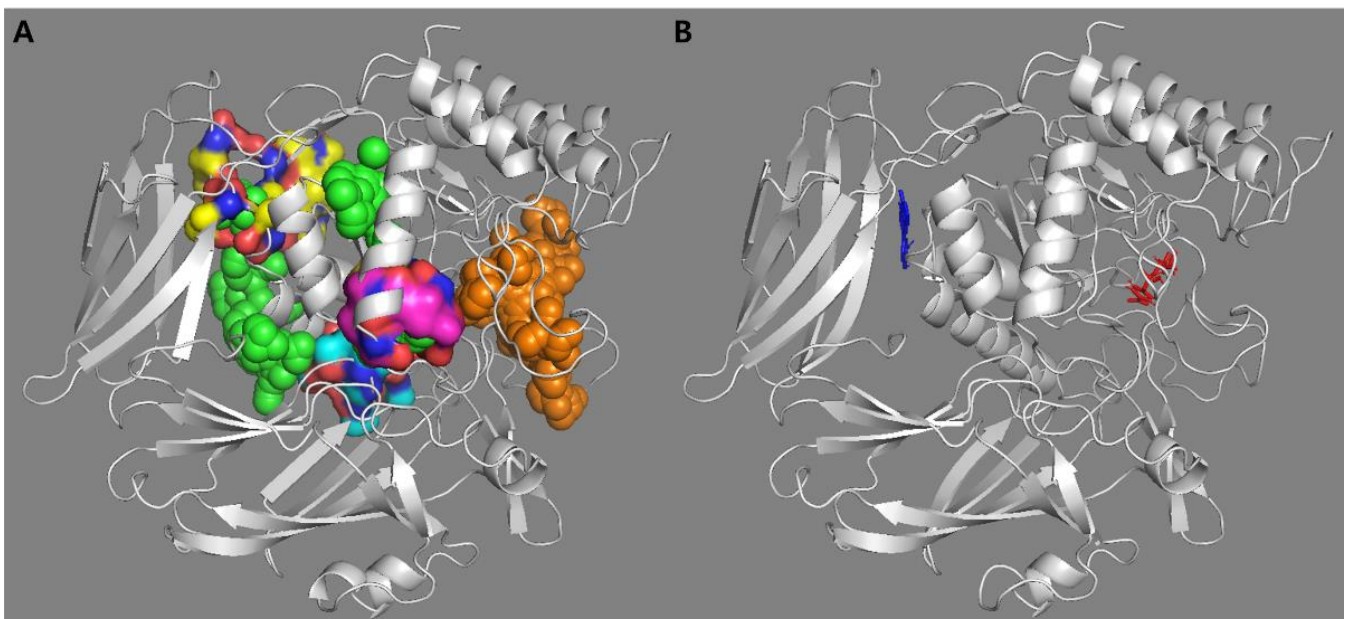

**Figure 4.** Predicted binding sites ((**A**); orange spheres: active site, green spheres: possible allosteric binding site predicted by AllositePro, and rainbow spheres: possible allosteric binding site predicted by PASSer2.0) and docking pose results of the substrate and compound **1** with the β-glucuronidase enzyme ((**B**); substrate: red; compound **1**: blue).

The docking results revealed that compound **1** could bind to the allosteric binding site of β-glucuronidase with a binding energy value of −8.35 kcal/mol (Figure 5 and Table 2). The carbonyl group and oxygen atom of **1** establish hydrogen bonding interactions with the amino acid residues, Thr191 and Thr438, whereas the hydroxy groups of **1** interact with Arg272, Leu435, and Asp436. Two benzene rings of **1** bind with residues Lys400 and Pro403 through π−alkyl interactions and Thr188 via π−σ interactions. The lactone ring displayed π−σ interactions with Val190, π−cation interactions with Arg439, and π−lone pair interactions with Val189. The other residues, including Gly270, Asn401, and Pro437, from different sites of the β-glucuronidase enzyme, bound to compound **1** via van der Waals interactions.

**Table 2.** Docking energies and binding site interactions of compound **1** with the β-glucuronidase enzyme.

| Comp. | Binding Energy (kcal/mol) | Hydrogen Bonds | van der Waals Interactions | Hydrophobic Interactions | Electrostatic Interactions | Others |
|---|---|---|---|---|---|---|
| **1** | −8.35 | Thr191 Arg272 Leu435 Thr438 | Gly270 Asn401 Pro437 | Thr188 (π−σ) Val190 (π−σ) Lys400 (π−alkyl) Pro403 (π−alkyl) | Arg439 (π−cation) | Val189 (π−lone pair) |

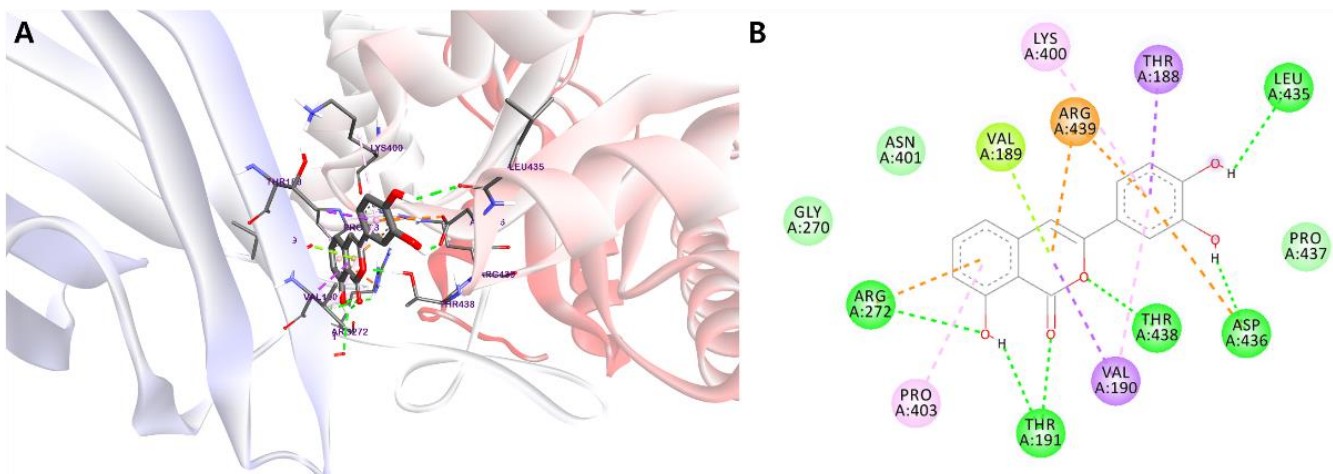

**Figure 5. The** 3D docking poses (**A**) and 2D interaction diagrams (**B**) of β-glucuronidase inhibition mediated by compound **1**. Green: hydrogen bonds, violet and pink: hydrophobic, orange: electrostatic, and light green: van der Waals interactions.

### 2.5. Molecular Dynamics Simulation of β-Glucuronidase Inhibition

To examine the structural stability and its variations of the **1**-6LEL complex, MD was performed after docking computations with a period of 100 ns, using the Desmond package (Schrödinger 2020-1, New York, NY, USA).

The general conditions of the simulation and its stabilization are obtained by root-mean-square deviation (RMSD) analysis [28]. The higher stability of the protein–ligand complex was demonstrated by a lower RMSD value, whereas decreased stability was indicated by an increased RMSD value [28]. The RMSD value of the **1**-6LEL complex increased rapidly during the initial equilibration fluctuation for 5 ns, slowly increasing from 5 to 38 ns, after which the RMSD was maintained between 2.2 and 2.4 Å, until the simulation was completed (Figure 6A). A maximum RMSD of 2.6 Å was observed for the target protein of β-glucuronidase, indicating that the **1**-6LEL complex maintained stability throughout the MD period. The flexibility and fluctuation of each residue in β-glucuronidase over the 100 ns simulation were represented by the root-mean-square fluctuation (RMSF) value used to predict the ligand binding-induced structural alterations in the protein structure [29]. Higher RMSF values represent greater flexibility in MD simulations [30]. The RMSF plot of the complex between compound **1** and β-glucuronidase is displayed in Figure 6B, where the peaks represent the β-glucuronidase regions that fluctuated the most during the 100 ns simulation. The RMSF values of amino acid residues in the allosteric site of β-glucuronidase were less than 1.3 Å. In contrast, the RMSF values of residues in the active site of β-glucuronidase fluctuate slightly more, between 2.0 and 3.5 Å, suggesting that the protein structure in ligand-bound conformations is stable during MD simulation. Hence, compound **1** may function as a non-competitive inhibitor of β-glucuronidase, which is consistent with the results of the kinetic study.

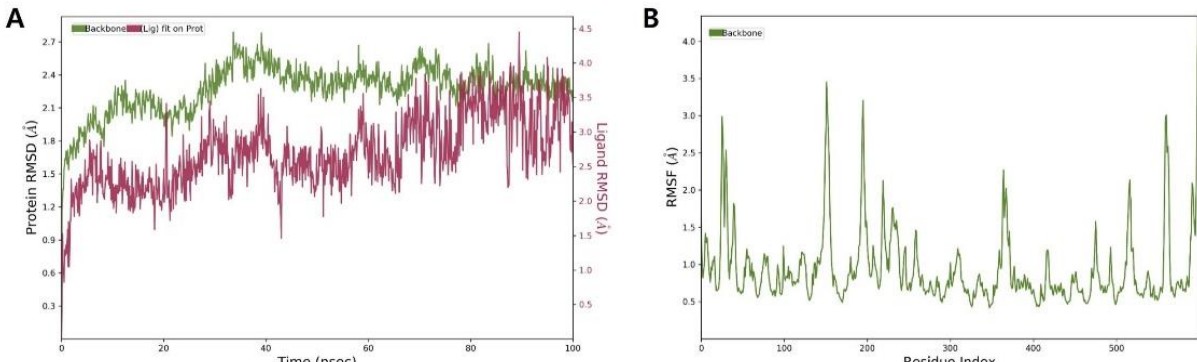

**Figure 6.** Molecular dynamics simulation of active compound **1** and *β*-glucuronidase (PDB ID: 6LEL) complex: RMSD ((**A**), protein RMSD: green line and RMSD of **1**: red line) and RMSF (**B**).

Figure 7 presents the protein–ligand contact diagram between compound **1** and *β*-glucuronidase. The hydroxy groups of **1** established hydrophobic contact with Leu435, polar interactions with Thr191 and His192, a positive charge with Lys400, and a negative charge with Asp436; these residues of the *β*-glucuronidase allosteric site contributed to the binding interactions of 24, 17, 13, 28, and 98%, respectively. In addition, the lactone ring of **1** was linked with Thr438 via polar interactions; Arg272 and Arg439 through π−cation interactions, accounting for 12 and 41% of the contribution, respectively. The MD results revealed the crucial role of the lactone ring in the *β*-glucuronidase inhibitory activity mediated by active compound **1**.

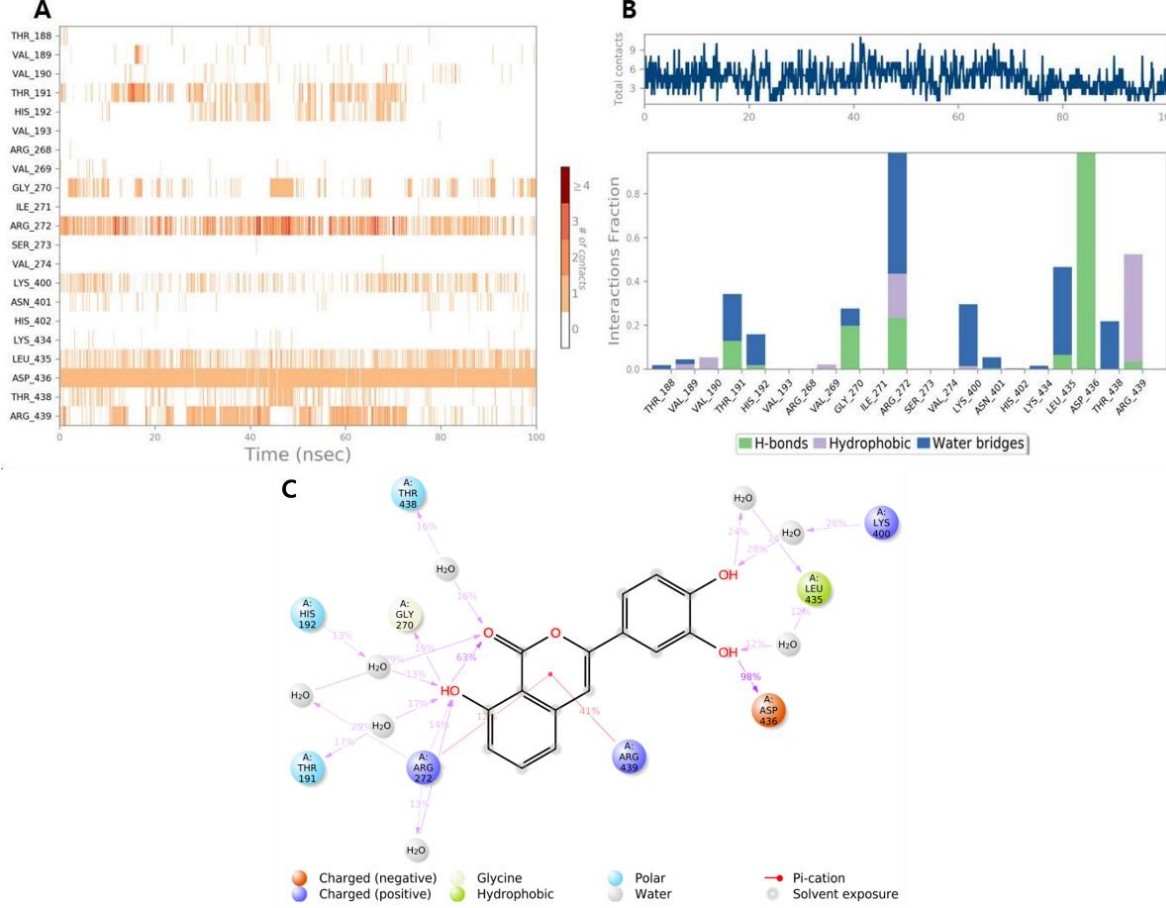

**Figure 7.** Protein–ligand contact analysis between compound **1** and *β*-glucuronidase complex ((**A**), timeline and (**B**), bar chart presentations) and 2D interaction diagram (**C**).

The ligand torsion plot that characterizes the conformational evolution of each rotatable bond (RB) in compound 1 during the 100 ns simulation trajectory is displayed in Figure 8A, where the 2D chemical structure of compound **1** with colored RB is followed by a same-colored dial and bar plots. A total of four RBs were observed in compound **1**, with the potential values of 3.36 kcal/mol and 7.16 kcal/mol for hydroxy groups and the linkage between the phenol ring and lactone ring, respectively.

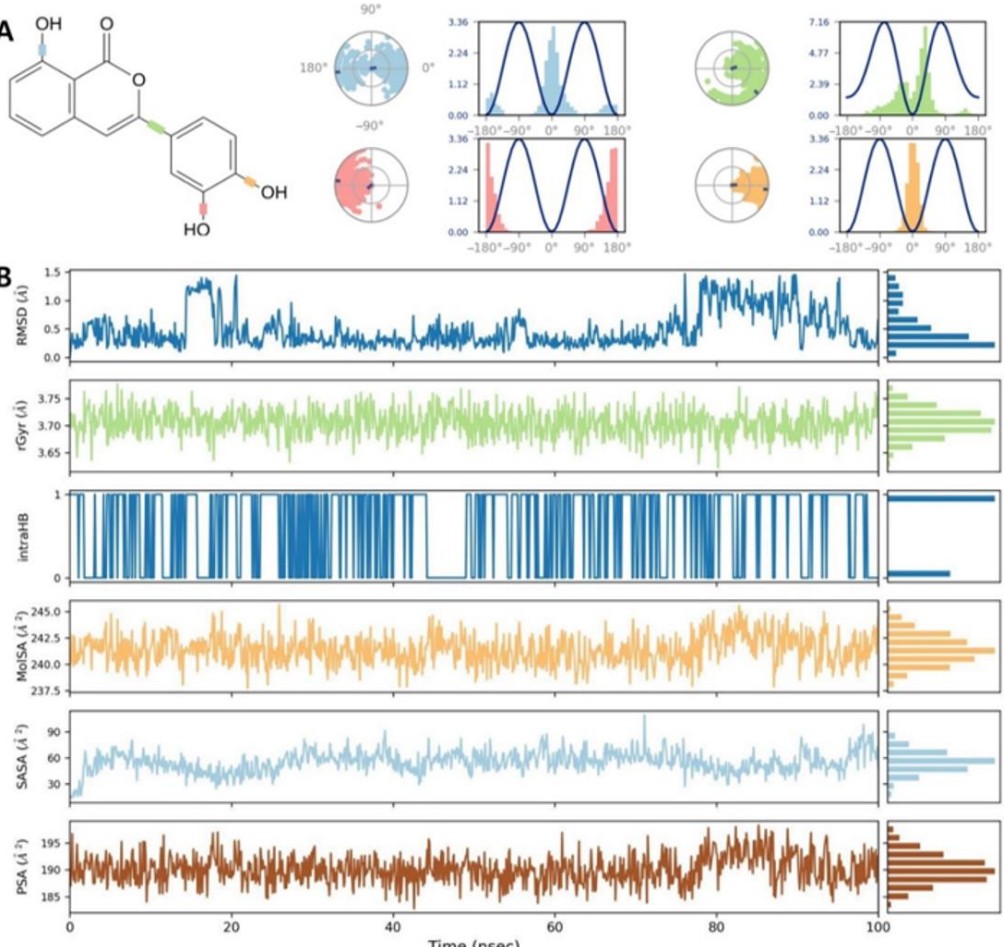

**Figure 8.** Ligand torsion profile (**A**) and ligand properties trajectory (**B**) of the **1**-6LEL complex throughout the simulation trajectory (0 to 100 ns).

Considering ligand properties, we evaluated the ligand RMSD, the radius of gyration (rGyr), intramolecular hydrogen bonds (intraHB), molecular surface area (MolSA), solvent-accessible surface area (SASA), and polar surface area (PSA) (Figure 8B). The RMSD of compound **1,** with respect to the reference conformation, ranged from 0.2 to 1.5 Å, and its balance was approximated at 0.45 Å. rGyr was analyzed to examine the stability of compound **1** in the allosteric binding site of *β*-glucuronidase during the 100 ns simulation. The **1**-6LEL complex exhibited an average rGyr value of 3.70 Å. No significant fluctuations were observed in rGyr, suggesting that the **1**-6LEL complex exhibited steady behavior. IntraHB refers to the number of internal hydrogen bonds within the ligand. The constant intraHB value for ligand **1** indicated the consistency of **1** during the simulation process. The MolSA value was calculated using a 1.4 Å probe radius and was equivalent to the van der Waals surface area. The MolSA value for ligand **1** was slightly altered, from 237.5 to 245.0 Å, throughout the 100 ns MD simulation. The SASA value represents the surface area of a molecule that can be accessed by a water molecule. The SASA value of the **1**-6LEL complex significantly increased from 25 to 60 Å until 4 ns of the MD simulation, followed by gradual stabilization at 56 Å. PSA is the SASA of a molecule, provided only by oxygen

and nitrogen atoms. The PSA of compound **1** fluctuated at a consistent rate throughout the 100 ns MD simulation, while the PSA value ranged between 185 and 195 Å.

## 3. Discussion

All the secondary metabolites were isolated from *A. mono* Maxim. (**1**−**8**) and evaluated to determine their β-glucuronidase inhibitory activity. Compound **1** exhibited a β-glucuronidase inhibitory effect, with an IC$_{50}$ value of 58.83 μM, whereas the remaining compounds demonstrated no inhibitory activity against β-glucuronidase. The results suggest that the different positions of an oxygen atom and carbonyl group in the lactone ring of the isocoumarin structure, as well as the presence of a double bond between C-3 and C-4, could contribute to β-glucuronidase inhibition. Kinetic analysis indicated that active compound **1** displayed non-competitive inhibition. Thus, molecular docking studies were employed to determine the binding position and critical amino acid interactions of compound **1** with the allosteric binding site of the β-glucuronidase protein. The docking results indicated that compound **1** could tightly bind to the allosteric binding site of β-glucuronidase, with a binding energy of −8.35 kcal/mol. The lactone ring of compound **1** displays hydrogen bonds with Thr191 and Thr438, π−σ interactions with Val190, π−cation interactions with Arg439, and π−lone pair interactions with Val189, which might explain their inhibitory activity against β-glucuronidase. In addition, the conformational stability of compound **1** complexed with the β-glucuronidase protein and its variations were investigated by performing MD simulation trajectories (100 ns). According to the MD results, all ligand properties fluctuated during the initial simulation period, gradually reaching equilibrium and a steady state as the simulation was completed. This indicated that compound **1** was stable in the allosteric binding site of β-glucuronidase. Based on the predicted pharmacokinetic properties (Supplementary Material), compound **1** exhibited no blood–brain barrier penetration and behaved as a safe drug candidate, given that it adhered to Lipinski's rule of five.

## 4. Materials and Methods

### 4.1. Experimental Procedures

$^1$H and $^{13}$C-NMR spectra were acquired on a Bruker Advance Digital 500 MHz instrument (Bruker, Karlsruhe, Germany). CC was conducted on silica gel 60 (230–400 mesh) and Cosmosil C$_{18}$ reversed phase gel (Nacalai Tesque, Kyoto, Japan). HPLC was conducted using a Waters HPLC system with a 2996 PDA detector (Waters, Milford, MA, USA). Thin-layer chromatography was conducted on pre-coated glass plates (silica gel 60 F$_{254}$ and RP-18 F$_{254s}$; Merck, Darmstadt, Germany). Plates were checked under ultraviolet radiation (254 and 365 nm), followed by spraying with sulfuric acid 10% and heating at 100–110 °C.

### 4.2. Chemicals and Reagents

HPLC solvents were supplied from Fisher Scientific Korea Ltd. (Seoul, Korea). *E. coli* β-glucuronidase enzyme (EC 3.2.1.31, G7396) and DSA (S0375) were provided by Sigma-Aldrich (St. Louis, MO, USA). PNPG (N0618) was provided by Tokyo Chemical Industry Co., Ltd. (Tokyo, Japan). All chemicals used in the experiments were provided by Duksan Pure Chemicals Inc. (Ansan, Gyeonggi, Korea).

### 4.3. Plant Material

Stems and branches of *A. mono* Maxim. were collected from the Medicinal Herb Garden of Daegu Catholic University in August 2021 and identified by Professor Byung Sun Min at the College of Pharmacy, Daegu Catholic University, Korea. A voucher specimen of *A. mono* Maxim. (23A-AM) was deposited at the Laboratory of Pharmacognosy, College of Pharmacy, Kyungpook National University, Korea.

### 4.4. Extraction and Isolation

Dried stems and branches of *A. mono* Maxim. (5.4 kg) were cut into small pieces and extracted with methanol (4 × 20 L) under reflux. The MeOH solvent was evaporated under reduced pressure to obtain the MeOH residue (350.0 g), which was subsequently suspended in $H_2O$ (3 L) and then partitioned with *n*-hexane, $CH_2Cl_2$, and EtOAc to obtain the *n*-hexane extract (35.2 g), $CH_2Cl_2$ extract (77.8 g), EtOAc extract (62.4 g), and water layer after removing the chemical solvents.

The $CH_2Cl_2$ extract was separated by CC on silica gel using a stepwise eluent of *n*-hexane–acetone (100:1–1:100, *v/v*) and $CH_2Cl_2$–MeOH (30:1–1:1, *v/v*) to yield seven fractions, 1A–G. Fraction 1D (8.1 g) was chromatographed on silica gel CC, eluted using a gradient mixture of $CH_2Cl_2$–acetone (100:1–1:100, *v/v*) and $CH_2Cl_2$–MeOH (20:1–1:1, *v/v*), to yield six subfractions, 1D1–6. Subfraction 1D2 (49.7 mg) was separated by RP-18 CC, using a mixture of acetone and $H_2O$ (1.5:1, *v/v*) as the eluent to obtain compound **3** (8.9 mg). Compounds **4** (5.8 mg) and **5** (2.0 mg) were purified from subfractions 1D3 (140.1 mg) and 1D4 (92.2 mg), respectively, by RP-18 CC elution with MeOH–$H_2O$ (1:1.5, *v/v*). Compounds **1** (4.9 mg) and **2** (8.9 mg) were isolated from subfraction 1D6 (223.2 mg) by HPLC using an isocratic mixture of MeOH and distilled $H_2O$ (40:60, *v/v*). Fraction 1E (9.5 g) was separated by silica gel CC, using $CH_2Cl_2$–acetone (gradient 100:1–1:100, *v/v*) and then $CH_2Cl_2$–MeOH (gradient 30:1–1:1, *v/v*) as the eluent to yield eight subfractions, 1E1−8. From subfraction 1E7 (239.2 mg), compounds **7** (44.5 mg) and **8** (17.4 mg) were isolated by HPLC, using 52% MeOH in distilled $H_2O$ as the eluent.

The water layer was chromatographed by Diaion CC using MeOH–$H_2O$ (gradient 0:1−1:0, *v/v*) as the eluent to yield three fractions, 2A−C. Fraction 2B (43.7 g) was chromatographed on silica gel and eluted with $CH_2Cl_2$–MeOH (gradient 100:1–1:100, *v/v*), followed by RP-18 CC using a MeOH–$H_2O$ mixture (1.5:1, *v/v*) as the mobile phase to obtain four fractions, 2B1−4. Compound **6** (7.2 mg) was isolated from fraction 2B1 (3.8 g) by HPLC, using an isocratic mixture of MeOH and $H_2O$ (50:50, *v/v*) as the eluent.

3-(3,4-Dihydroxyphenyl)-8-hydroxyisocoumarin (**1**), yellow needles, [1]H-NMR (500 MHz, $CD_3OD-d_4$): $\delta_H$ 7.56 (1H, t, *J* = 7.9 Hz, H-6), 7.46 (1H, d, *J* = 2.0 Hz, H-2′), 7.32 (1H, d, *J* = 7.6 Hz, H-5), 7.11 (1H, dd, *J* = 8.2, 2.0 Hz, H-6′), 6.87 (1H, d, *J* = 8.1 Hz, H-7), 6.80 (1H, d, *J* = 7.6 Hz, H-5′), 6.46 (1H, s, H-4); [13]C-NMR (125 MHz, $CD_3OD-d_4$): $\delta_C$ 167.9 (C-1), 144.1 (C-3), 108.6 (C-4), 116.6 (C-5), 137.8 (C-6), 116.4 (C-7), 158.3 (C-8), 109.9 (C-9), 143.8 (C-10), 126.9 (C-1′), 111.7 (C-2′), 146.5 (C-3′), 147.5 (C-4′), 117.7 (C-5′), 124.2 (C-6′).

### 4.5. β-Glucuronidase Inhibition Assay

The β-glucuronidase inhibition assay was evaluated as previously described [7].

### 4.6. β-Glucuronidase Kinetics Assay

The β-glucuronidase kinetics assay was performed as previously described [7].

### 4.7. Molecular Docking

Docking simulations were conducted using AutoDock 4.2, following our previously described protocol [7]. The crystallographic structure of β-glucuronidase was retrieved from the RCSB PDB website (PDB ID: 6LEL) [31]. The allosteric site of β-glucuronidase was predicted and generated using the AllositePro method provided by the AlloSite 2.10 web server and PASSer2.0 protein allosteric sites server [26,27].

### 4.8. Molecular Dynamics Simulation

MD simulations were performed using the Desmond package (Schrödinger 2020-1, New York, NY, USA). The protein–ligand complex was prepared in a 10.0 × 10.0 × 10.0 Å orthorhombic box (simple point-charge solvation model) [32]. Next, a 0.15 M NaCl solution and counter-ions were added to the system for neutralization. The solvated system was energy-minimized, and its position was restrained with the OPLS3e force field. The minimized system was implemented in an NPT ensemble at 300 K and 1 atm. Finally, the

MD simulation was conducted to run for 100 ns, and 1000 frames were generated, with a recording interval of 100 ps.

### 4.9. Statistics

All results are expressed as the mean ± standard error (SEM) of the three independent experiments. Statistical significance was analyzed using one-way analysis of variance (ANOVA) and Duncan's test (*p*-value < 0.05).

### 5. Conclusions

In the present study, we performed a chemical investigation of the stems and branches of *A. mono* Maxim., which resulted in the purification and structural elucidation of eight known compounds (two isocoumarins and six coumarins). To the best of our knowledge, our study is the first report that isolated isocoumarins from an *Acer* species. In addition, we, for the first time, determined the inhibitory activity of the isolated molecules against *β*-glucuronidase. Our results revealed that 3-(3,4-dihydroxyphenyl)-8-hydroxyisocoumarin (**1**) inhibited *β*-glucuronidase activity. The results of the kinetic analysis were consistent with the molecular docking studies, suggesting that compound **1** could bind to the *β*-glucuronidase allosteric site. This result was supported by MD studies up to 100 ns, which confirmed the binding stability of the protein–ligand complex in the trajectory analysis. These findings imply that the complex of *β*-glucuronidase and compound **1** is quite stable in biological systems. Moreover, the pharmacokinetic properties of compound **1** were analyzed, and the results suggested that compound **1** could be a promising drug candidate, given that no violations of the drug-likeness rules were observed.

**Supplementary Materials:** The following supporting information can be downloaded at: https://www.mdpi.com/article/10.3390/app122010685/s1, Figure S1. Bioavailability radar of compound **1**; Table S1. Drug-likeness properties of compound **1** [33–37].

**Author Contributions:** Conceptualization and methodology, S.Y.Y.; investigation, and data curation, N.V.P.; formal analysis, writing—original draft preparation, N.V.P. and S.Y.Y.; writing—review and editing, B.S.M. and J.A.K.; supervision, J.A.K. All authors have read and agreed to the published version of the manuscript.

**Funding:** This research was supported by the National Research Foundation of Korea (NRF) grant funded by the Korean government (MSIT) (No. NRF-2020R1A5A2017323 and NRF-2022R1C1C1004636).

**Institutional Review Board Statement:** Not applicable.

**Informed Consent Statement:** Not applicable.

**Data Availability Statement:** Not applicable.

**Conflicts of Interest:** The authors declare no conflict of interest.

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
