# Peer review of "Inhibitory Effect of Coumarins and Isocoumarins Isolated from the Stems and Branches of Acer mono Maxim. against Escherichia coli β-Glucuronidase"

_applsci, doi:10.3390/app122010685_

Round 1

Reviewer 1 Report

The authors made a good job of isolating, identifying and evaluating the biological properties of the obtained compounds.

Therefore, in my opinion, the results presented do warrant publication in this journal after corrections:

1. Line 107. Authors affirm that distinct positions of an oxygen atom and ketone play a significant role in mediating the β-glucuronidase inhibitory effect, but there are only 2 type of positions of these substituent, as well as the presence/absence of a double bond. Moreover, there are a lot of type of substituent, which can play role in the presence of the activity. So, I am not sure that there is a place to be SAR definition in relation to oxygen. Thus, the authors need to either revise the sentence or add the supplement conclusions about SAR experiment. The same remark applies to the sentence on line 271-272

2. Line 122. Under table 1 the «2Positive control» should be more specific (definition), or need some information about it in the part 2.2.

Author Response

The authors made a good job of isolating, identifying and evaluating the biological properties of the obtained compounds.

Answer: The authors are thankful to the reviewer for your time reviewing our manuscript and your insightful comments. The changes that we have made following the Reviewer’s comments in the revised manuscript are highlighted in yellow color.

Therefore, in my opinion, the results presented do warrant publication in this journal after corrections:

  1. Line 107. Authors affirm that distinct positions of an oxygen atom and ketone play a significant role in mediating the β-glucuronidase inhibitory effect, but there are only 2 type of positions of these substituent, as well as the presence/absence of a double bond. Moreover, there are a lot of type of substituent, which can play role in the presence of the activity. So, I am not sure that there is a place to be SAR definition in relation to oxygen. Thus, the authors need to either revise the sentence or add the supplement conclusions about SAR experiment. The same remark applies to the sentence on line 271-272

Answer: The authors would like to thank the Reviewer for your valuable comment. We fully agree with the Reviewer that the SAR conclusion of these isolated compounds need more experiments. Thus, the section about the SAR analysis (Section 2.2 and lines 271-272) was revised in the revised manuscript.

  1. Line 122. Under table 1 the «2Positive control» should be more specific (definition), or need some information about it in the part 2.2.

Answer: As suggested by the Reviewer, the information of the positive control (DSA) was added in the section 2.2 of the revised manuscript.

Reviewer 2 Report

This manuscript presents the isolation of 8 known compounds from the stems and branches of a tree of the Acer genus.

All compounds were tested for activity as inhibitors of E. Coli b-glucuronidase. Only one compound showed activity and was further studied by docking and molecular dynamics studies.

The results indicate that this active compound may be a candidate for further studies.

This is an interesting manuscript. However, there are a series of issues that must be addressed before it can be considered for publication.

First, in the Abstract, it says that those compounds were isolated for the first time, and although the authors probably meant the first time from this particular tree, a reader may be induced to believe that those are new compounds. What should be indicated is that this is the first time that isocoumarins are isolated from this genus.

All compounds are known, and the only one active in the biological tests conducted has been reported in more than 40 papers, 33 of them regarding its biological activity. The rest of the compounds isolated are inactive in the test presented here

The active compound has been named Thunberginol A and can be checked on Wikipedia. Since it is a well-known molecule with different activities studied, the ADMET and Lipinski studies are probably redundant.

The inhibition kinetics, docking, and molecular dynamic studies are pertinent and give interest to the manuscript.

Thus, the Abstract and manuscript should be written in a way that makes clear from the beginning that the reported compounds are known and that the goal of the manuscript is to study this biological activity (inhibition of E. Coli b-glucuronidase).

The use of abbreviations should be explained the first time they are used (i.e. DSA in Figure 2 and Table 1)

A lactone is not composed of oxygen and a ketone, but of oxygen and a carbonyl group. A ketone requires two alkyl or aryl groups.

In the kinetic studies, it is concluded that the Ki value is 40.9 uM, but no context is given as to what this value means. It should be indicated.

Author Response

This manuscript presents the isolation of 8 known compounds from the stems and branches of a tree of the Acer genus.

All compounds were tested for activity as inhibitors of E. Coli b-glucuronidase. Only one compound showed activity and was further studied by docking and molecular dynamics studies.

The results indicate that this active compound may be a candidate for further studies.

This is an interesting manuscript. However, there are a series of issues that must be addressed before it can be considered for publication.

Answer: The authors are thankful to the reviewer for your time reviewing our manuscript and your insightful comments. The changes that we have made following the Reviewer’s comments in the revised manuscript are highlighted in yellow color.

First, in the Abstract, it says that those compounds were isolated for the first time, and although the authors probably meant the first time from this particular tree, a reader may be induced to believe that those are new compounds. What should be indicated is that this is the first time that isocoumarins are isolated from this genus.

Answer: The authors fully agree with the Reviewer that the Abstract may cause misunderstanding to the reader that isolated compounds is new ones. Thus, the Abstract section was rewritten to make clear to the reader.

All compounds are known, and the only one active in the biological tests conducted has been reported in more than 40 papers, 33 of them regarding its biological activity. The rest of the compounds isolated are inactive in the test presented here

The active compound has been named Thunberginol A and can be checked on Wikipedia. Since it is a well-known molecule with different activities studied, the ADMET and Lipinski studies are probably redundant.

Answer: The authors really appreciate your comments. The active compound 1 in our study is a known compound and reportedly exhibited various pharmacological effects (33 papers). However, our study is the first report on the β-glucuronidase inhibitory effect and mechanism of this compound. Also, drug-likeness properties of this compound have not been reported in all published papers. Therefore, the authors really hope to provide readers the information about the pharmacokinetics properties of compound 1 and other models can be further conducted without worrying about the safety of the compound.

However, we fully agree with the Reviewer that compound 1 is a well-known molecule with different activities studied. Thus, we suggest moving the drug-likeness properties discussion part (section 2.5) to the Supplemental material.

The inhibition kinetics, docking, and molecular dynamic studies are pertinent and give interest to the manuscript.

Thus, the Abstract and manuscript should be written in a way that makes clear from the beginning that the reported compounds are known and that the goal of the manuscript is to study this biological activity (inhibition of E. Coli b-glucuronidase).

Answer: The authors are thankful for the valuable comment of the reviewer. As suggested by the Reviewer, the Abstract and Conclusions section was revised to make clear that the isolated flavonoids are known compounds and the goal of this study is the investigation of β-glucuronidase inhibitory activity.

The use of abbreviations should be explained the first time they are used (i.e. DSA in Figure 2 and Table 1)

Answer: As suggested by the Reviewer, the explanation of DSA was added in the section 2.2 of the revised manuscript.

A lactone is not composed of oxygen and a ketone, but of oxygen and a carbonyl group. A ketone requires two alkyl or aryl groups.

Answer: The authors would like to thank the Reviewer for your valuable comments. We fully agree with the Reviewer that a lactone ring is composed of oxygen and a ketone group. As suggested by the Reviewer, this information about the lactone was changed in the revised manuscript.

In the kinetic studies, it is concluded that the Ki value is 40.9 uM, but no context is given as to what this value means. It should be indicated.

Answer: As suggested by the Reviewer, the meaning of Ki value was added in the section 2.2 of the revised manuscript.

Round 2

Reviewer 2 Report

After the changes introduced by the authors, the manuscript is now suitable for publication